# Is serotonin transporter brain binding associated with the cortisol awakening response? An independent non-replication

**Juliane Conradi**[1☉], **Jonas E. Svensson**[1☉], **Søren V. Larsen**[1,2], **Vibe G. Frokjaer**[1,2,3]*

**1** Neurobiology Research Unit, Copenhagen University Hospital, Rigshospitalet, Copenhagen, Denmark,
**2** Department of Clinical Medicine, Faculty of Health and Medical Sciences, University of Copenhagen, Copenhagen, Denmark, **3** Mental Health Services Capital Region Denmark, Psychiatric Center Copenhagen, Copenhagen, Denmark

☉ These authors contributed equally to this work.
* vibe.frokjaer@nru.dk

## Abstract

### Background

Serotonergic brain signaling is considered critical for an appropriate and dynamic adaptation to stress, seemingly through modulating limbic system functions, such as the hypothalamic-pituitary-adrenal (HPA)-axis. This interplay is of great interest since it holds promise as a target for preventing stress-related brain disorders, e.g., major depression. Our group has previously observed that prefrontal serotonin transporter (5-HTT) binding, imaged with positron emission tomography (PET), is positively associated with the cortisol awakening response (CAR), an index of HPA axis stress hormone dynamics. The aim of this cross-sectional study was to replicate the previous finding in a larger independent group of healthy individuals.

### Methods

Molecular imaging and cortisol data were available for 90 healthy individuals. Prefrontal 5-HTT binding was imaged with [$^{11}$C]DASB brain PET. Non-displaceable 5-HTT binding potential (BP$_{ND}$) was quantified using the Multilinear Reference Tissue Model 2 (MRTM2) with cerebellum as the reference region. CAR was based on five serial salivary cortisol samples within the first hour upon awakening. The association between CAR and prefrontal 5-HTT BP$_{ND}$ was evaluated using a multiple linear regression model adjusted for age and sex. Further, we tested for sex differences in the association. Finally, an exploratory analysis of the association, was performed in 8 additional brain regions.

### Results

We observed no statistically significant association between 5-HTT binding and CAR corrected for age and sex in the prefrontal cortex ($\beta$ = -0.28, p = 0.26). We saw no interaction with sex on the association (p = 0.99).

**Data Availability Statement:** Due to the General Data Protection Regulation (GDPR), the data that support the findings of this study are not readily available. However, data in the Cimbi database can

be accessed by application (http://www.cimbi.dk/db).

**Funding:** The financial support for this study's database infrastructure was provided by The Research Council of Rigshospitalet (https://www.rigshospitalet.dk/research) in the form of a grant (A6594) awarded to VF, JC, JS, and SL. Additional financial support for this study's database infrastructure was provided by The Lundbeck Foundation (https://lundbeckfonden.com/en) in the form of grants (279-2018-1145 and R90-A7722) awarded to VF, JC, JS, and SVL. This study was also financially supported by "Ivan Nielsens fond for personer med specielle sindslidelser" (https://www.legatbogen.dk/ivan-nielsens-fond-for-personer-med-specielle-sindslidelser) in the form of a grant (R162-A7167) awarded to JC. This study was also financially supported by Independent Research Fund Denmark (https://dff.dk/en) in the form of a grant (7025-00111B) awarded to JC, and grants (0134-00278B and 7025-00111B) awarded to SVL. This study was also financially supported by Capital Region's Research Foundation for Health Research (https://research.regionh.dk/en/) in the form of a grant (R162-A7167) awarded to JS. This study was also financially supported by The Swedish Brain Foundation in the form of a grant awarded to JS. The funders had no role in study design, data collection and analysis, decision to publish, or preparation of the manuscript.

**Competing interests:** I have read the journal's policy and the authors of this manuscript have the following competing interests: VGF have previously received honorarium as a consultant for Sage Therapeutics and for lectures for Lundbeck Pharma A/S, Gedeon-Richter and Janssen-Cilag A/S. This does not alter our adherence to PLOS ONE policies on sharing data and materials. None of the authors has an ongoing affiliation or employment with a commercial company or other for-profit organization.

## Conclusion

We could *not* confirm a positive association between CAR and prefrontal 5-HTT $BP_{ND}$ in this independent dataset. Also, sex differences in the association were not apparent. Our data do not exclude that the serotonin transporter system is involved in the regulation of stress responses in at-risk or manifest depressed states.

## Introduction

The serotonin (5-HT) system is involved in a multitude of psychophysiological functions e.g., appetite, reward responses, cognition, memory, mood, sleep, and stress responses [1]. The 5-HT transporter (5-HTT) is the primary mechanism to regulate extracellular 5-HT and is the target of the most commonly used antidepressants, which block 5-HTT [2]. Cerebral 5-HT signaling has been proposed to mediate stress resilience through control of limbic system functions, including the hypothalamic–pituitary–adrenal (HPA) axis, the major neuroendocrine system that governs stress responses [3, 4]. Similarly, the serotonin system is sensitive to HPA-axis mediated glucocorticoid alterations resulting from stressful stimuli [5]. A better understanding of the interaction between the cerebral 5-HT system and HPA-axis could open new avenues for preventing, diagnosing, and stratifying the treatment of stress-related psychiatric conditions, such as depression and generalized anxiety disorder.

The interplay between HPA-axis activity and cerebral 5-HTT can be studied in vivo through the cortisol awakening response (CAR), which reflects stimulated HPA-axis activity [6], and in vivo molecular imaging of markers of 5-HT brain signaling using Positron emission tomography (PET). In a previous study, we have shown that CAR is positively associated with prefrontal 5-HTT binding in healthy individuals [7], this study is henceforth referred to as "the original study". A similar association was observed in a cohort of participants with recent but not acute use of MDMA [8]. These cross-sectional results align with the notion that pre-frontal 5-HT neurotransmission may modulate CAR. Also, later work of ours using other PET markers i.e., imaging of the 5-HT$_4$ receptor (5-HT$_4$R) binding show that 5-HT$_4$R binding is negatively associated with the magnitude of CAR [9]. Under the assumption that 5-HT$_4$R binding is an inverse marker of serotonergic tone [10], this reconciles with the notion that 5-HT signaling modulates cortisol dynamics. (i.e., higher 5-HT tone more dynamics in terms of higher CAR). Thus, 5-HT signaling may support a balanced stress hormone response and possibly helps navigate natural stressors in a healthy manner.

A limitation of the studies described above is that they were based on relatively small study groups with a skewed sex distribution. Frokjaer, Erritzoe et al. 2013 [7], studied a group with N = 32, of whom 7 were women and Frokjaer, Erritzoe et al. 2014 [8], a group of N = 18 with recent but not acute use of MDMA, of whom 2 were women. Small sample sizes are a common problem in research fields with resource demanding experiments, like brain PET. This increases the risk of unreliable results which cannot be reproduced [11]. It is therefore important to evaluate if earlier findings replicate using independent samples to assess published results' robustness and adjust the interpretations if relevant.

Here we aim to test our previous findings of a positive association between prefrontal 5-HTT binding and CAR in a larger, independent dataset, with a higher proportion of women and a PET-system with higher resolution compared with the original study.

## Materials and methods

The participants in both groups (i.e., original and replication group) were healthy controls in studies all registered and approved by the local ethics committee and conducted in accordance with the Declaration of Helsinki. Written informed consent was obtained from the participants prior to participation. Data allowing for identification of individual participants was available to the researchers during data analysis, no data from medical records was accessed for this study. The Center for Integrated Molecular Brain Imaging (Cimbi) [12] database stores the largest collection of high-resolution 5-HT neuroimaging data available from healthy individuals worldwide. The database was established in 2008 and data collection is still ongoing. The original study group consisted of 32 healthy controls who participated in previous clinical studies accessible within the Cimbi database structure [12]. For the replication group, we included participants with available 5-HTT PET scan and CAR data reflecting HPA-axis activity, collected after the original study was performed, using the Cimbi database [12]. Excluding participants in the original study, the combination of data on CAR and 5-HTT binding was available for 102 healthy individuals. In line with the original study, participants taking steroid hormone substitution or psychoactive medication were not included, nor were participants with more than one month between PET scan and CAR sampling or those with insufficient CAR data due to non-compliance to sampling instructions. Only three participants had been scanned with the older GE-Advance PET system. In order to reduce variance, a decision was made, prior to initiation of data analysis, not to include these participants in the analysis. Consequently, in total, 90 healthy participants (67 women) between 18–48 years of age (25.3±6, mean±SD) were available for the independent replication analysis. As a description of perceived stress, Cohen's perceived stress scale was used [13]. See results section for a comparison of clinical profiles from both the replication- and original study populations.

### Magnetic resonance imaging and PET experimental procedure

T1-weighted magnetic resonance (MR) images were acquired using a 3T Siemens Magnetom Trio or Verio MR scanner. All participants were examined using a HRRT PET scanner (Siemens Molecular Imaging, USA), with an approximate in-plane spatial resolution of 2mm [14]. A six-minute transmission scan was performed prior to each PET measurement to correct for signal attenuation. The radioligand [11C]DASB, which binds selectively to 5-HTT, was prepared as described earlier [15]. In each PET experiment, a saline solution containing [11C]DASB was injected into an antecubital vein as a bolus. The cannula was then flushed with saline. Emission data were acquired continuously for 90 min and subsequently binned into 36 consecutive time frames using the following frame definitions: six 10 s, three 20 s, six 30 s, five 1 min, five 2 min, eight 5 min, and three 10 min frames.

### Regions of interest

In the original study, prefrontal [11C]DASB binding was positively associated with the cortisol response to awakening. In an exploratory, voxel based, analysis a highly significant positive association was identified in a cluster of voxels centred in Brodmann's area 25, which approximately corresponds to the subcallosal gyrus. Hence, for the replication study, the prefrontal cortex, and subcallosal gyrus were considered in the analysis. Two separate methods were applied to create the regions of interest (ROIs). The MR image for each participant was processed as follows: 1) through the software suit FreeSurfer (version 7.2, http://surfer.nmr.mgh.harvard), using the Destrieux atlas, the subcallosal gyrus was delineated; 2) Using the same approach as in the original analysis, the Pvelab software package [16] was used to delineate prefrontal cortex and cerebellar cortex, which is used as a reference when quantifying the

5-HTT binding. The following 8 ROIs where the [11C]DASB signal to noise ratio was deemed acceptable were included in a post-hoc, exploratory analysis: amygdala, anterior cingulate gyrus (ACC), posterior cingulate gyrus (PCC), caudate, hippocampus, insular cortex, putamen, and thalamus.

## Image analysis and quantification of 5-HTT

The AIR algorithm [17] was used to motion correct the dynamic PET images and co-register a summated PET image with the MR image from the same individual. The co-registration was confirmed by visual inspection in all planes. Using the co-registration matrix, the ROIs were then projected on the dynamic PET image to derive regional time-activity curves (TACs). To obtain non-displaceable binding potential ($BP_{ND}$) estimates, kinetic modelling of the TACs was performed for each region, using the reference tissue MRTM2 model [18]. Cerebellar grey matter (excluding Vermis) was used as reference region. In the original analysis, partial volume correction (PVC) using the Mueller-Gartner approach was applied [7]. Since the replication group was examined using a PET system with higher resolution, no such correction was applied in this study.

## The cortisol awakening response (CAR)

CAR, measured as the area under the curve with respect to increase from awakening levels (AUCi) reflects the dynamic of post-awakening increase in salivary cortisol and the subsequent restoration over the following hour [19]. AUCi was based on five serial salivary home samples taken (0, 15, 30, 45, and 60 min) after awakening. For details on the sampling and analysis procedures, see [7]. The cortisol home samples were made between -6 to 25 days from the PET scan date (1.3±3.1). Sampling was initiated no later than 10 minutes after awakening, with only four participants reporting >5 minutes from awakening to sample initiation. The timespan in which all the samples were taken varied from 55 to 71 minutes (60.7±2.8). All AUCi measures were normalized to a 60-minute timespan in the original study and the same procedure was applied in the present analysis.

## Preregistration and analysis plan

Following recent recommendations for PET studies [20], the image analysis and statistical plan were preregistered at AsPredicted.com (See supporting information, S1 Text). The preregistration was submitted before image analysis for the current project was performed but after data collection was completed due to the database nature of the study.

The analysis plan was as follows: To reanalyse the original study group to test if a positive association between CAR and 5-HTT binding could be found using a subcallosal gyri ROI. In case of a positive association, this ROI would then be used in the primary replication analysis. However, if there was no positive finding, the plan was to use the same delineation of brain ROIs as in the original study (i.e., the prefrontal cortex) for the replication analysis.

## Statistics

For our analyses of the original group, a multiple linear regression model was used to test the association between CAR and 5-HTT binding in the subcallosal gyrus and prefrontal cortex while adjusting for age. For the analysis in the replication group, we used a multiple linear regression model, adjusting for age and sex, with the rationale that both variables may influence CAR [6] and 5-HTT $BP_{ND}$ [21]. The statistical test was one-sided, expecting a positive association between the variables.

Given an observed difference in Cohen's perceived stress score and BMI between the original and replication sample we evaluated the effect of adjusting for these variables in the model. We also investigated the effect of sex on the association between [$^{11}$C]DASB binding and CAR by adding an interaction term to the model.

To visualize the association between the variables after adjusting for age and sex, the residuals from two separate regression equations were plotted against each other after adding the means of the raw variables to the residuals [22].

For the primary analysis a p-value below 0.05 was considered significant. For the exploratory analysis no pre-specified hypothesis was tested, therefore we did not apply the conventional null hypothesis significance testing, where an alpha is set for error control [23]. Since no claims regarding error rate was made, no correction for multiple comparisons was performed.

For all other reported tests, unless otherwise specified, Welch's two-sample t-test was used to test for difference between groups for continuous variables and the chi-squared test for categorical data.

R version 4.1.3 was used to perform graphical presentations and statistical analyses.

## Results

Measures of CAR and prefrontal 5-HTT $BP_{ND}$ and descriptive information on the study groups are shown in Table 1. Compared with the original study, the replication group had a higher proportion of women (22% and 74%, respectively, p<0.001) and was, on average ten years younger (p = 0.01). The original group, on average, had a lower CAR than the replication group (p = 0.03) medians of 41 and 183 respectively, a lower perceived stress score (p<0.01) and a higher BMI (p<0.03).

### Association between 5-HTT binding and CAR

In a reanalysis of the data from the original study, we reproduced the results of a positive association between prefrontal 5-HTT binding and CAR adjusted for age (β = 1.5, p = 0.02). The

**Table 1. Descriptive information on study groups.**

| | Replication study, n = 90 | Original study, n = 32 | Test |
|---|---|---|---|
| Parameter | Mean±SD (range) | Mean±SD (range) | p-value |
| Age (years) | 25.3 ± 6.2 (18.4–47.7) | 35.3 ± 20.1 (19.7–81.7) | 0.01 |
| Sex, female | 74% (67 of 90) | 22% (7 of 32) | <0.01 |
| BMI (kg/m$^2$) | 23.5 ± 3.6 (17.4–43.1) | 25.0 ± 3.2 (17.9–32.9) | 0.03 |
| Alcohol (units per week) | 7.7 ± 12.6 (0–100) | 8.6 ± 6.9 (0–25) | 0.61 |
| Smoking | 4.4% (4 of 90) | 12.5% (4 of 32) | 0.22[a] |
| Smoking, package years | 3.8 ± 11.2 (0–52) | 3.5 ± 4.8 (0–17) | 0.89 |
| Prefrontal 5-HTT $BP_{ND}$ | 0.4 ± 0.067 (0.27–0.58) | 0.58 ± 0.079 (0.44–0.82)* | <0.01 |
| Cohen's perceived stress score | 8.4±6.0 (0–23) | 5.2±3.5 (0–14)** | <0.01 (0.02[b]) |
| CAR, AUCi$_{timecorrected}$ (nmol/L/min) | 174 ± 287 (-471-1095) | 59.5 ± 230 (-446-588) | 0.03 (0.03[b]) |

[a]For the categorical variable; Smoking, Fisher's exact test was used.

[b] Wilcoxon signed rank test.

Abbreviations: AUCi, Area under the curve with respect to increase from awakening; BMI, Body mass index; $BP_{ND}$, Non-displaceable binding potential; CAR, Cortisol awakening response.

*Partial volume corrected $BP_{ND}$ values.

**n = 27. The average Cohen's perceived stress score differs from the one reported in the original study (4.7±3.6). Since source raw data were used here, this discrepancy reflects an error in the original calculations.

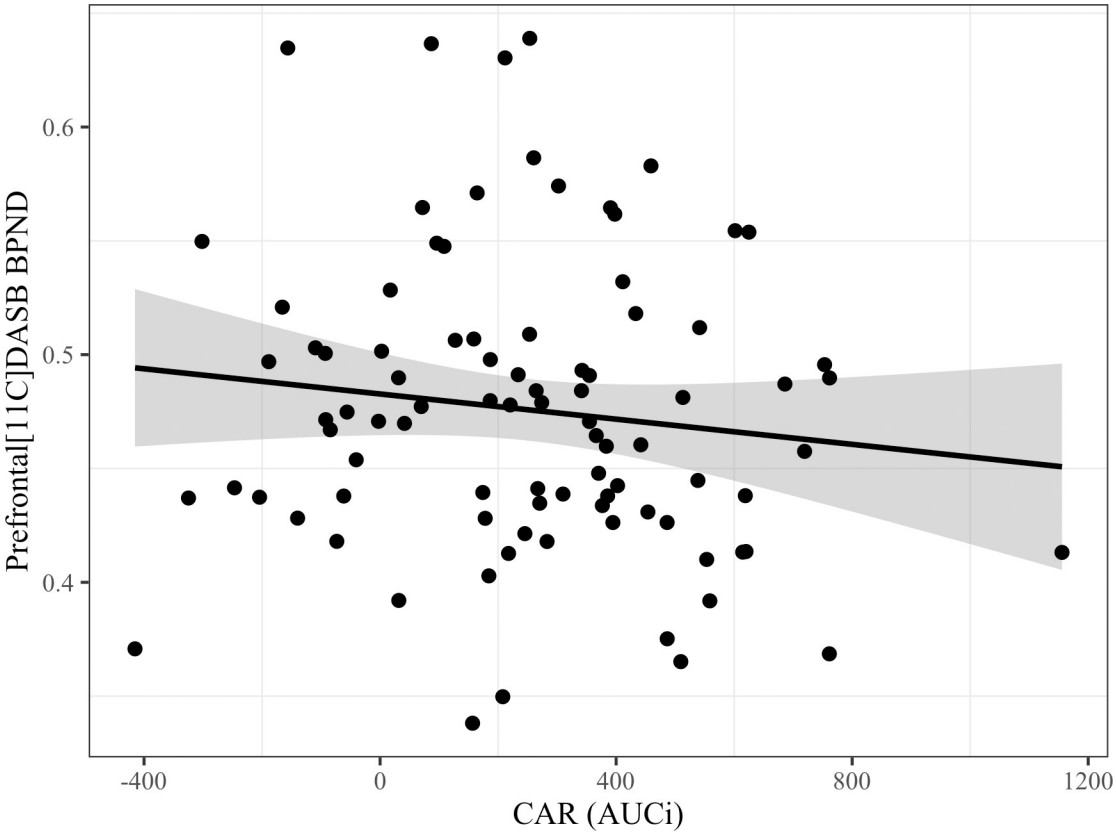

**Fig 1. Scatter plot depicting the non-significant association between prefrontal 5-HTT binding and CAR after adjusting for age and sex.** Fitted regression line in black, with the shaded grey area representing the 95% confidence interval, p-value = 0.26. Abbreviations: AUCi, Area under the curve with respect to increase from awakening; $BP_{ND}$, Non-displaceable binding potential; CAR, cortisol awakening response.

beta denotes the effect of CAR on 5-HTT $BP_{ND}$ being $BP_{ND}$ per $10^{-13}$ mol cortisol/L/min. We observed no statistically significant association between 5-HTT binding and CAR corrected for age in the subcallosal gyrus ($\beta = 0.28$, $p = 0.78$) as modelled in FreeSurfer using the Destrieux atlas. In accordance with the analysis plan, we, therefore, used prefrontal cortex binding estimates for the primary replication analysis. In the analysis of the replication group, we did not find a significant association between prefrontal 5-HTT binding and CAR, adjusting for age and sex ($\beta = -0.28$, $p = 0.26$) (Fig 1). We observed no significant interaction effect of sex on the age-adjusted association between prefrontal 5-HTT binding and CAR; the slope-estimate for the men differed insignificantly by $\beta = -0.00023$ ($p = 0.99$) compared to the women.

The replication group scored higher on Cohen's perceived stress scale compared to the original group (medians of 7.5 and 5.2 respectively). In order to test whether the degree of perceived stress or BMI affected the association, we added these variables as covariates in the model. The association between prefrontal 5-HTT binding and CAR, adjusting for age, sex and Cohen's perceived stress score, was similar to the primary analysis ($\beta = -0.27$, $p = 0.26$), as were the results when adjusting for BMI ($\beta = -0.28$, $p = 0.25$).

In an exploratory analysis we tested the association between CAR and 5-HTT binding, adjusted for age and sex, in 8 additional brain regions (See supporting information, S1 Table). In all regions the association was negative, consistent with the result in prefrontal cortex. The strongest effect was observed in the caudate nuclei ($\beta = -2.4$, $p = 0.04$).

## Discussion

In this study we have evaluated if our previously reported finding of a positive association between CAR and 5-HTT binding in prefrontal cortex [7] replicates in an independent population of healthy adults. We could not confirm a positive association between CAR and prefrontal 5-HTT binding using a larger study group of primarily women and a PET system with higher resolution.

### Non-replication

There are several possible explanations for why we did not replicate our earlier finding. First, the present study was performed in a group of primarily young women, while participants in the original study was mostly men of a wider age range. CAR dynamics have been shown to differ between men and women [24]. This could explain the differences in average CAR between the two groups and affect an association with 5-HTT, especially since HPA-axis activity is affected by hormonal status in women [25]. However, the replication analysis was controlled for both age and sex, and also an interaction analysis did not indicate that the association between CAR and 5-HTT was dependent on sex. Therefore, we consider this explanation less likely. Second, the study groups differed significantly in $[^{11}C]DASB$ $BP_{ND}$. This is expected, given that a PET-camera system of higher resolution was used in the replication study, and that partial volume effect correction was performed in the original study. This difference in magnitude of $BP_{ND}$ should in theory have no consequence on the results since the rank order of subjects should be the same in both cameras. Third, 15 out of 32 participants in the original study were recruited as healthy controls in a study on recent but not acute hallucinogenic or MDMA use [26]. The recruitment of the control group was done to match a study group using recreational drugs such as MDMA, which could contribute to differences between the participants in the original- and replication study. For instance, the two groups differed significantly on average Cohen's perceived stress scores, where the replication group scored higher and had a lower average BMI. In a sensitivity analysis, however, adding these variables as covariates did not change the association's direction, effect size, or significance level. Histograms illustrating the frequency distribution of CAR and Cohen's PSS for the two study groups, did not reveal any clear outliers driving the between-group differences for these parameters (see Supporting information, Figures A-D in S1 Appendix). The group differences on key variables is a potential problem for this study, however, we would expect a robust and generalizable underlying association to appear regardless of these differences. Nevertheless, we cannot exclude that the study groups differed on other parameters, not registered, with potential relevance for the CAR or 5-HTT settings or the relation between these. Given the above differences between the original sample and the replication sample this study cannot be considered a *direct* replication attempt, but can be viewed as a conceptual replication [27, 28].

Fourth, another explanation for the non-significant finding in this replication effort is that there is no direct coupling between prefrontal 5-HTT binding and cortisol dynamics, or that it is not possible to reliably detect such coupling using $[^{11}C]DASB$ PET and CAR. Failures to replicate findings are common in science [29], and there is no reason to expect otherwise in the neuroimaging field, where sample sizes are often small and power low [11]. However, since an association is also seen in former users of MDMA we cannot exclude that in specific more extreme subgroups 5-HTT may be linked to CAR [8].

### Cortisol dynamics and serotonin signaling in high-risk or depressed groups

The coupling between 5-HT signaling and cortisol dynamics has been demonstrated in several previous studies by our group [7–9]. Interestingly, in the MDMA user group [8] an association

between CAR and prefrontal 5-HTT binding similar to the control group, i.e., from the original study was observed, suggesting, that in the off-balance state of recent recreational MDMA use, the serotonergic system and HPA-axis are coupled in a detectable manner. Using other measures, Steinberg et al. has shown a positive association between cortisol responses to the Trier Social Stress Test and brain 5-HT$_{1A}$ binding [30], alterations of which has been linked to stress-related disorders like MDD and suicidal behavior [31]. Similarly, Reimold et al. 2011 [32] saw a negative association between the dexamethasone-corticotropin challenge and thalamic 5-HTT binding in patients with OCD, unipolar depression, and healthy controls. It is relevant to consider changes in 5-HT levels or 5-HT release capacity as part of a coping mechanism to stress, which may promote healthy adaption to adverse events [33, 34]. Moreover, the seemingly disturbed serotonin release capacity in patients with MDD [35] can possibly lead to maladaptation to stress and either trigger or maintain a depressed state. An interpretation of this is, in high-risk or depressed states, cortisol dynamics and serotonin signaling may interact differently than in euthymic states.

## Perspectives

Our exploratory analysis of the independent dataset showed negative associations between 5-HTT binding and CAR in all examined brain regions, with the strongest association in the caudate nuclei ($\beta$ = -2.4, p = 0.04). Given that this was an exploratory analysis on a relatively large number of ROIs, without correction for multiple comparisons, this result should be interpreted with utmost caution but could lend some support for future studies focusing on regional specificity.

Given the inconsistency of findings regarding the association between 5-HTT and CAR, further investigation of the interplay between the serotonin system and HPA axis dynamics in vivo is needed. One way to move forward could be to apply longitudinal study designs, and map differences between healthy states, at-risk states e.g., familial risk for depression, burnout, manifest MDD and suicide attempters [36, 37].

## Methodological considerations

This study has some methodological limitations: i) The initial analysis, using the subcallosal gyrus ROI showed no significant association with CAR, this in contrast with the voxel-based analysis conducted in the original study. A possible explanation for this is that the subcallosal ROI is relatively small and thus with a poor signal-to-noise ratio that does not allow us to capture an underlying association. ii) Alternative cortisol measures, such as the dexamethasone challenge, could be considered to assess inhibitory feedback loops in the HPA-axis dynamics which may rely on serotonin signaling. However, CAR measurement is a non-invasive procedure done in a natural environment without the potential confounding influence of stress from instrumental procedures in a laboratory [38]. iii) Interindividual variation in violations of the assumptions underlying reference tissue modelling of PET data can introduce variance in the binding estimates which may attenuate a true underlying correlation [39].

## Conclusion

In this replication attempt, we could not confirm a positive association between CAR and prefrontal 5-HTT BP$_{ND}$ in healthy adults. Also, sex differences in the association were not apparent. Our data do not exclude that the serotonin system is involved in the regulation of stress responses in off-balance, at-risk or manifest depressed states.

## Supporting information

**S1 Table. Average 5-HTT BP$_{ND}$ and linear regression results for association between CAR and 5-HTT BP$_{ND}$.**
(PDF)

**S1 Appendix. Histograms illustrating distributions of CAR and Cohen's perceived stress scores in the two study groups.** Figures A-D.
(PDF)

**S1 Text. Preregistration.**
(PDF)

## Acknowledgments

The authors would like to thank the staff at the Neurobiology Research Unit who have been involved in the continuous collection and storing of data in the Cimbi database. Also, we are grateful to all study participants.

## Author Contributions

**Conceptualization:** Juliane Conradi, Jonas E. Svensson, Søren V. Larsen, Vibe G. Frokjaer.

**Data curation:** Juliane Conradi, Jonas E. Svensson, Vibe G. Frokjaer.

**Formal analysis:** Juliane Conradi, Jonas E. Svensson, Søren V. Larsen, Vibe G. Frokjaer.

**Funding acquisition:** Vibe G. Frokjaer.

**Investigation:** Søren V. Larsen, Vibe G. Frokjaer.

**Methodology:** Juliane Conradi, Søren V. Larsen, Vibe G. Frokjaer.

**Project administration:** Juliane Conradi, Jonas E. Svensson, Søren V. Larsen, Vibe G. Frokjaer.

**Resources:** Vibe G. Frokjaer.

**Supervision:** Jonas E. Svensson, Søren V. Larsen, Vibe G. Frokjaer.

**Validation:** Vibe G. Frokjaer.

**Visualization:** Juliane Conradi, Jonas E. Svensson, Søren V. Larsen, Vibe G. Frokjaer.

**Writing – original draft:** Juliane Conradi.

**Writing – review & editing:** Jonas E. Svensson, Søren V. Larsen, Vibe G. Frokjaer.

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
