## [Decision Letter · Decision Letter 0]

29 May 2023

PONE-D-23-06771Is serotonin transporter brain binding associated with the cortisol awakening response? An independent non-replicationPLOS ONE

Dear Dr. Gedsø Frøkjær,

Thank you for submitting your manuscript to PLOS ONE. After careful consideration, we feel that it has merit but does not fully meet PLOS ONE’s publication criteria as it currently stands. Therefore, we invite you to submit a revised version of the manuscript that addresses the points raised during the review process.

We look forward to receiving your revised manuscript.

Kind regards,

Hidenori Yamasue, M.D., Ph.D.

Academic Editor

PLOS ONE

Journal Requirements:

2. In the Methods section and the online submission form, please provide additional information about the patient records used in your retrospective study. Specifically, please ensure that you have discussed whether all data were fully anonymized before you accessed them and/or whether the IRB or ethics committee waived the requirement for informed consent. If patients provided informed written consent to have data from their medical records used in research, please include this information.

“Database infrastructure was supported by The Research Council of Rigshospitalet, grantID A6594 and The Lundbeck Foundation, grantIDs R279-2018-1145 and R90-A7722. JC was supported by “Ivan Nielsens fond for personer med specielle sindslidelser”, grantID 07018017 and the Independent Research Fund Denmark, grantID 7025-00111B. JS was supported by a grant from Capital Region's Research Foundation for Health Research, grantID R162-A7167. SVL was supported by the Independent Research Fund Denmark, grantID 0134-00278B and 7025-00111B.”

“I have read the journal's policy and the authors of this manuscript have the following competing interests: VGF has received honorarium as a consultant for Sage Therapeutics and lectures for Lundbeck Pharma A/S and Janssen-Cilag A/S.”

We note that one or more of the authors are employed by a commercial company: Sage Therapeutics and lectures for Lundbeck Pharma A/S and Janssen-Cilag A/S

Reviewers' comments:

Reviewer's Responses to Questions

**Comments to the Author**

1. Is the manuscript technically sound, and do the data support the conclusions?

Reviewer #1: Partly

Reviewer #2: Yes

2. Has the statistical analysis been performed appropriately and rigorously? 

Reviewer #1: I Don't Know

Reviewer #2: Yes

3. Have the authors made all data underlying the findings in their manuscript fully available?

Reviewer #1: Yes

Reviewer #2: Yes

4. Is the manuscript presented in an intelligible fashion and written in standard English?

Reviewer #1: Yes

Reviewer #2: Yes

5. Review Comments to the Author

Reviewer #1: The authors report failing to replicate a previously reported finding from this group and awakening cortisol response correlates positively with PFC 5-HTT binding. There are several aspects of this study that seem glossed over. First the two study samples were scanned with different PET scanner and the 5-HTT binding levels differ significantly. The cortisol levels and reported perceived stress levels also differ between the two samples. Even in the first sample the correlations was present in only one of the two brain regions examined. So there are many ways by which the study samples differ including on the two measures that they are seeking a correlation for in this study. They have an odd idea that high transporter binding means lower serotonin release but in fact it is intra-synaptic cortisol that regulates HTT internalization and so when there is less serotonin release this increase internalization of 5-HTT and reduces the number of transporters.

There are technical issues. They do not employ an input function and so factors like brain blood flow or variable nonspecific binding could add noise and they cannot detect that. They could present data on more ROIs from the original sample as well as the replication sample and see how widespread or narrow the original finding really was and the same for the replication sample. Finally they can pout their CAR response in perspective by looking at other PET studies of serotonin markers and cortisol like Steinberg et al and Bartlett et al from Columbia group.

Reviewer #2: This manuscript attempts to replicate the authors' prior findings of a relationship between serotonin transporter or 5HTT measured using [11C]DASB PET binding and cortisol awakening response (CAR), an index of HPA axis stress hormone dynamics, in healthy adults. The authors observed that prefrontal 5-HTT binding is positively associated with CAR. The authors were not able to replicate their prior findings in a sample of 90 healthy adults.This is an important research question to understand the role of HPA axis dynamics in serotonin availability. Replicating a prior finding is also very important given the limited reproducibility of findings. However, there were several concerns about the manuscript.

1- The manuscript was not clear on the original sample size and the nature of the sample. This manuscript needs to be evaluated without the need to go to the published findings and some of the study design details of the original sample are important to present here. This is especially important as it seemed from the discussion that the original sample was a healthy sample.

2. The definition of blunted HPA axis activity by using <50% increase in CAR is not justified. The authors has already analyzed this variable continuously. Without a rationale that is well justified, this definition is arbitrary.

3. The sample sizes in the table comparing the samples need to be presented. There were differences between the samples on BMI but the authors did not control for BMI.

4. The manuscript does not acknowledge work from the Columbia group (Dr. John Mann) where they used this ligand in depressed samples.

6. PLOS authors have the option to publish the peer review history of their article (what does this mean?). If published, this will include your full peer review and any attached files.

Reviewer #1: No

Reviewer #2: No

---

## [Author Response · Author response to Decision Letter 0]

22 Jun 2023

Response to Reviewer comments

We thank the reviewers for valuable comments on our manuscript Is serotonin transporter brain binding associated with the cortisol awakening response? An independent non-replication (MS Number: PONE-D-23-06771). We have amended the text according to the reviewer’s comments and believe that the manuscript has improved as a result. 

Below follows a point-by-point response, first to the comments by, the academic editor, thereafter to the comments by the reviewers. Our responses are written (in roman) to each point raised (in italics). 

Points raised by the academic editor

We have updated the manuscript to meet PLOS ONE’s style requirements using the recommended style templates. See the ‘Manuscript’ file. 

2. In the Methods section and the online submission form, please provide additional information about the patient records used in your retrospective study. Specifically, please ensure that you have discussed whether all data were fully anonymized before you accessed them and/or whether the IRB or ethics committee waived the requirement for informed consent. If patients provided informed written consent to have data from their medical records used in research, please include this information.

The participants in both the original and replication study were healthy controls in previous clinical studies accessible upon application within the Cimbi database structure. We have updated this information in the methods section, see paragraph 1. 

"Database infrastructure was supported by The Research Council of Rigshospitalet, grantID A6594 and The Lundbeck Foundation, grantIDs R279-2018-1145 and R90-A7722. JC was supported by "Ivan Nielsens fond for personer med specielle sindslidelser", grantID 07018017 and the Independent Research Fund Denmark, grantID 7025-00111B. JS was supported by a grant from Capital Region's Research Foundation for Health Research, grantID R162-A7167. SVL was supported by the Independent Research Fund Denmark, grantID 0134-00278B and 7025-00111B."

Please provide an amended statement that declares *all* the funding or sources of support (whether external or internal to your organization) received during this study, as detailed online in our guide for authors at http://journals.plos.org/plosone/s/submit-now. Please also include the statement "There was no additional external funding received for this study." in your updated Funding Statement.

Please see the updated funding statement in ‘Cover letter’.

4. We note that the grant information you provided in the 'Funding Information' and 'Financial Disclosure' sections do not match.

When you resubmit, please ensure that you provide the correct grant numbers for the awards you received for your study in the 'Funding Information' section.

We have ensured that the correct grant information is provided in the ‘funding information’ section in the submission system.

"I have read the journal's policy and the authors of this manuscript have the following competing interests: VGF has received honorarium as a consultant for Sage Therapeutics and lectures for Lundbeck Pharma A/S and Janssen-Cilag A/S."

We note that one or more of the authors are employed by a commercial company: Sage Therapeutics and lectures for Lundbeck Pharma A/S and Janssen-Cilag A/S

"The funder provided support in the form of salaries for authors [insert relevant initials], but did not have any additional role in the study design, data collection and analysis, decision to publish, or preparation of the manuscript. The specific roles of these authors are articulated in the 'author contributions' section."

Within your Competing Interests Statement, please confirm that this commercial affiliation does not alter your adherence to all PLOS ONE policies on sharing data and materials by including the following statement: "This does not alter our adherence to PLOS ONE policies on sharing data and materials." (as detailed online in our guide for authors http://journals.plos.org/plosone/s/competing-interests). If this adherence statement is not accurate and there are restrictions on sharing of data and/or materials, please state these. Please note that we cannot proceed with consideration of your article until this information has been declared.

Thank you for your guidance on how to update the competing interests and funding statement to meet PLOS ONE requirements. See ‘Cover letter’ for updated sections. We would like to make it clear that none of the authors have any employment or affiliation with any commercial organization. We have updated the funding statement stating the role of the funders.

We have updated the figures to fit your requirements using the PACE tool. The figure file is inserted in the manuscript and uploaded as a separate file in the submission system. 

 

Points raised by the reviwers

Reviewer #1: The authors report failing to replicate a previously reported finding from this group and awakening cortisol response correlates positively with PFC 5-HTT binding. There are several aspects of this study that seem glossed over. 

1) First the two study samples were scanned with different PET scanner and the 5-HTT binding levels differ significantly. The cortisol levels and reported perceived stress levels also differ between the two samples. Even in the first sample the correlations was present in only one of the two brain regions examined. So there are many ways by which the study samples differ including on the two measures that they are seeking a correlation for in this study. 

There are indeed differences in key variables between the original sample and the replication sample. We have now expanded on this discussion in the manuscript (see the ‘Non-replication’ paragraph in the discussion section). The camera system used in the replication sample has a higher resolution, which, through reduced partial volumes effects, will lead to different binding estimates than lower-resolution systems. This is a well-described phenomenon, and while it can make it difficult to directly translate binding estimates between experiments on different systems, the rank order between subjects, and thus any association with other measurements (e.g., CAR data) should be the same regardless of camera. 

Therefore, given a robust and generalizable underlying association between the examined variables, we would expect it to be detected regardless of these differences

2) They have an odd idea that high transporter binding means lower serotonin release but in fact it is intra-synaptic cortisol that regulates HTT internalization and so when there is less serotonin release this increase internalization of 5-HTT and reduces the number of transporters.

Thank you for highlighting this section of the text which indeed is speculative and not something that our data can directly inform on. We have now revised the manuscript accordingly (Introduction, second paragraph).

3) There are technical issues. They do not employ an input function and so factors like brain blood flow or variable nonspecific binding could add noise and they cannot detect that.

We agree that the use of reference tissue modeling for quantification in neuro PET comes with certain limitations and can contribute to noise. A sentence regarding this has been added to the discussion section (paragraph under the header “Methodological considerations”). For [11C]DASB, the radioligand used in this study, reference tissue modeling has been common practice for more than two decades[1]. Notably, the same method is applied in the replication study as in the original study.

4) They could present data on more ROIs from the original sample as well as the replication sample and see how widespread or narrow the original finding really was and the same for the replication sample.

In the original study, the association between CAR and [11C]DASB binding was reported in the prefrontal cortex and the anterior cingulate cortex (ACC). In both regions, a positive association was observed, but statistically significant only in the prefrontal cortex. Previously we had not reported on ACC in the replication sample but have now added this analysis to the manuscript. The association was negative and not significant (see methods section under “Regions of interest” and results section under “Association between 5-HTT binding and CAR”). 

The primary objective of the current study is to attempt a replication of the significant finding in the original study. Although we see value in a larger exploratory analysis beyond this, such an approach comes with a risk of new spurious findings, undermining the purpose of the replication attempt. Beyond adding ACC to the analysis of the replication sample we are hesitant to add further analyses to this study.

5) Finally they can put their CAR response in perspective by looking at other PET studies of serotonin markers and cortisol like Steinberg et al and Bartlett et al from Columbia group.

The work on this topic by the Columbia group is indeed central and the lack of incorporation of this in the manuscript was noted by both reviewers. We have now added central references in the appropriate sections of the manuscript (see discussion section under heading 3 and 4.)

Reviewer #2: This manuscript attempts to replicate the authors' prior findings of a relationship between serotonin transporter or 5HTT measured using [11C]DASB PET binding and cortisol awakening response (CAR), an index of HPA axis stress hormone dynamics, in healthy adults. The authors observed that prefrontal 5-HTT binding is positively associated with CAR. The authors were not able to replicate their prior findings in a sample of 90 healthy adults.This is an important research question to understand the role of HPA axis dynamics in serotonin availability. Replicating a prior finding is also very important given the limited reproducibility of findings. However, there were several concerns about the manuscript.

1) The manuscript was not clear on the original sample size and the nature of the sample. This manuscript needs to be evaluated without the need to go to the published findings and some of the study design details of the original sample are important to present here. This is especially important as it seemed from the discussion that the original sample was a healthy sample.

Thank you for pointing this out, we have now clarified the information about the cohort studied in the original study. See updated table 1 and methods section, first paragraph. 

2) The definition of blunted HPA axis activity by using <50% increase in CAR is not justified. The authors has already analyzed this variable continuously. Without a rationale that is well justified, this definition is arbitrary.

Thank you, upon reconsideration we agree with this point. This analysis has now been removed from the manuscript. 

3) The sample sizes in the table comparing the samples need to be presented. There were differences between the samples on BMI but the authors did not control for BMI.

The table is now updated. We have run the analysis with BMI as a covariate. This did not change the results notably. This analysis is reported in the last paragraph of the results section.

4) The manuscript does not acknowledge work from the Columbia group (Dr. John Mann) where they used this ligand in depressed samples.

The work on this topic by the Columbia group is indeed central and the lack of incorporation of this in the manuscript was noted by both reviewers. We have now added central references in the discussion section under heading 3 and 4.)

References:

1. Ichise M, Liow JS, Lu JQ, Takano A, Model K, Toyama H, et al. Linearized reference tissue parametric imaging methods: application to [11C]DASB positron emission tomography studies of the serotonin transporter in human brain. J Cereb Blood Flow Metab. 2003;23(9):1096-112. Epub 2003/09/16. doi: 10.1097/01.WCB.0000085441.37552.CA. PubMed PMID: 12973026.

---

## [Decision Letter · Decision Letter 1]

20 Jul 2023

PONE-D-23-06771R1Is serotonin transporter brain binding associated with the cortisol awakening response? An independent non-replicationPLOS ONE

Dear Dr. Gedsø Frøkjær,

Thank you for submitting your manuscript to PLOS ONE. After careful consideration, we feel that it has merit but does not fully meet PLOS ONE’s publication criteria as it currently stands. Therefore, we invite you to submit a revised version of the manuscript that addresses the points raised during the review process.

We look forward to receiving your revised manuscript.

Kind regards,

Hidenori Yamasue, M.D., Ph.D.

Academic Editor

PLOS ONE

Journal Requirements:

**Additional Editor Comments:**

The authors revised the manuscript in response to the reviewer's commnets on the initial submission adequately. Now, as one reviewer additionally pointed out one issue, the authors should respond to the comment.

Reviewers' comments:

Reviewer's Responses to Questions

**Comments to the Author**

1. If the authors have adequately addressed your comments raised in a previous round of review and you feel that this manuscript is now acceptable for publication, you may indicate that here to bypass the “Comments to the Author” section, enter your conflict of interest statement in the “Confidential to Editor” section, and submit your "Accept" recommendation.

Reviewer #2: (No Response)

2. Is the manuscript technically sound, and do the data support the conclusions?

Reviewer #2: Yes

3. Has the statistical analysis been performed appropriately and rigorously? 

Reviewer #2: I Don't Know

4. Have the authors made all data underlying the findings in their manuscript fully available?

Reviewer #2: No

5. Is the manuscript presented in an intelligible fashion and written in standard English?

Reviewer #2: Yes

6. Review Comments to the Author

Reviewer #2: The authors have addressed my comments and other reviewers' comments. However, one remaining concern is the differences between the original sample and replication samples on perceived stress and CAR. There might be some outliers on these measures that are driving these differences. More details about this is needed.

In addition, given the differences between the samples, the authors need to tone down the replication perspective of this work. This is not strictly a replication and it is a missed opportunity not to look at other brain regions given the larger sample of the "replication sample". The original sample is small and could have resulted in a spurious association that cannot be replicated.

7. PLOS authors have the option to publish the peer review history of their article (what does this mean?). If published, this will include your full peer review and any attached files.

Reviewer #2: No

---

## [Author Response · Author response to Decision Letter 1]

11 Aug 2023

Response to Reviewer comments

We thank the reviewers for valuable comments on our manuscript Is serotonin transporter brain binding associated with the cortisol awakening response? An independent non-replication (MS Number: PONE-D-23-06771R1). We have amended the manuscript accordingly to the reviewer’s comments and believe that the manuscript has further improved. 

Below follows a point-by-point response to the comments by the reviewers. Our responses are written (in roman) to each point raised (in italics). 

Points raised by reviewer #2

Reviewer point 1: However, one remaining concern is the differences between the original sample and replication samples on perceived stress and CAR. There might be some outliers on these measures that are driving these differences. More details about this is needed.

We agree that the differences between the original and replication sample on central parameters is an important limitation for the replication study. In order to assess whether the observed difference could be driven by the presence of outliers in our data we have performed the following analyses:

1) For both CAR and perceived stress score we are now reporting both mean and median values in the results section. No large differences indicative of skewed distributions were observed.

2) Histograms for both variables have been attached as supplementary figures. No clear outliers can be readily identified in these figures.

3) The differences between groups remain significant when applying the Wilcoxon signed rank test, which is less sensitive to outliers compared with the t-test (p=0.03 and p=0.02 for the difference between CAR and Cohen’s PSS respectively).

We have attempted to outline possible explanations for the difference between the original and replication group in CAR and Cohen’s PSS:

“There are several possible explanations for why we did not replicate our earlier finding. First, the present study was performed in a group of primarily young women, while participants in the original study was mostly men of a wider age range. CAR dynamics have been shown to differ between men and women [1]. This could explain the differences in average CAR between the two groups and affect an association with 5-HTT, especially since HPA-axis activity is affected by hormonal status in women [2]. However, the replication analysis was controlled for both age and sex, and also an interaction analysis did not indicate that the association between CAR and 5-HTT was dependent on sex. Therefore, we consider this explanation less likely.”

And 

“The recruitment of the control group was done to match a study group using recreational drugs such as MDMA, which could contribute to differences between the participants in the original- and replication study. For instance, the two groups differed significantly on average Cohen’s perceived stress scores, where the replication group scored higher and had a lower average BMI. In a sensitivity analysis, however, adding these variables as covariates did not change the association’s direction, effect size, or significance level. Histograms illustrating the frequency distribution of CAR and Cohen’s PSS for the two study groups, did not reveal any clear outliers driving the between-group differences for these parameters (see Supporting information, Figure A-D in S1 Appendix.)

Reviewer point 2: In addition, given the differences between the samples, the authors need to tone down the replication perspective of this work.

We agree with the reviewer that this effort to replicate an earlier finding is not strictly a direct replication of the original study due to differences in key variables. We are now referring to this work as a conceptual replication in the manuscript:

Given the above differences between the original sample and the replication sample this study cannot be considered a direct replication attempt, but can be viewed as a conceptual replication[3, 4].

We have toned down the wording regarding replication throughout the manuscript:

- We omitted ‘replicate’ from this sentence in introduction section:

“Here we aim to test our previous findings of a positive association between prefrontal 5-HTT binding and CAR in a larger, independent dataset, with a higher proportion of women and a PET-system with higher resolution compared with the original study.”

- We replaced ‘replicate’ with ‘find’ in this sentence of the methods section:

“In the analysis of the replication group, we did not find a significant association between prefrontal 5-HTT binding and CAR, adjusting for age and sex (β= -0.28, p=0.26) (Fig 1).”

- In the discussion we previously propose that the non-replication could be explained by the original finding being a type I error. However, such statement is too strong given that the present work cannot qualify as a direct replication. Instead we now write: 

“Fourth, another explanation for the non-significant finding in this replication effort is that there is no direct coupling between prefrontal 5-HTT binding and cortisol dynamics, or that it is not possible to reliably detect such coupling using [11C]DASB PET and CAR.”

- In the conclusion we changed the wording from ‘independent replication study’ to ‘replication attempt’ in the following sentence:

“In this replication attempt, we could not confirm a positive association between CAR and prefrontal 5-HTT BPND in healthy adults.”

Reviewer point 3: This is not strictly a replication and it is a missed opportunity not to look at other brain regions given the larger sample of the "replication sample". The original sample is small and could have resulted in a spurious association that cannot be replicated.

We have now as supplementary material provided additional exploratory analyses in 8 brain regions with a relatively high density of serotonin transporters (Anterior cingulate cortex, putamen, caudate nuclei, thalamus, insula, hippocampus, amygdala, posterior cingulate cortex). All observed associations were negative (i.e., in the opposite direction compared with the original finding), this is expected given the high correlation between brain regions for the in vivo concentration of most examined proteins related to the serotonin system [5]. To note, the analysis showed a p-value of 0.04 in one region (caudate nucleus), though this result would not survive an adjustment for multiple comparisons, it could warrant future attention.

We have added this to the ’Perspectives’ paragraph of the discussion section:

“Our exploratory analysis of the independent dataset showed negative associations between 5-HTT binding and CAR in all examined brain regions, with the strongest association in the caudate nuclei (β = -2.4, p=0.04). Given that this was an exploratory analysis on a relatively large number of ROIs, without correction for multiple comparisons, this result should be interpreted with utmost caution but could lend some support for future studies focusing on regional specificity.”

1. Nasser A, Ozenne B, Hogsted ES, Jensen PS, Frokjaer VG. Reliability of three versus five saliva sampling times for assessing the cortisol awakening response. Psychoneuroendocrinology. 2022;147:105950. Epub 2022/10/23. doi: 10.1016/j.psyneuen.2022.105950. PubMed PMID: 36272363.

2. Gervasio J, Zheng S, Skrotzki C, Pachete A. The effect of oral contraceptive use on cortisol reactivity to the Trier Social Stress Test: A meta-analysis. Psychoneuroendocrinology. 2022;136:105626. Epub 2021/12/19. doi: 10.1016/j.psyneuen.2021.105626. PubMed PMID: 34922094.

3. Zwaan RA, Etz A, Lucas RE, Donnellan MB. Making replication mainstream. Behav Brain Sci. 2017;41:e120. Epub 2017/10/27. doi: 10.1017/S0140525X17001972. PubMed PMID: 29065933.

4. Nosek BA, Errington TM. Making sense of replications. Elife. 2017;6. Epub 2017/01/20. doi: 10.7554/eLife.23383. PubMed PMID: 28100398; PubMed Central PMCID: PMCPMC5245957.

5. Beliveau V, Ozenne B, Strother S, Greve DN, Svarer C, Knudsen GM, et al. The structure of the serotonin system: A PET imaging study. Neuroimage. 2020;205:116240. Epub 2019/10/11. doi: 10.1016/j.neuroimage.2019.116240. PubMed PMID: 31600591; PubMed Central PMCID: PMCPMC6951807.

---

## [Editor Report · Decision Letter 2]

13 Aug 2023

Is serotonin transporter brain binding associated with the cortisol awakening response? An independent non-replication

PONE-D-23-06771R2

Dear Dr. Gedsø Frøkjær,

We’re pleased to inform you that your manuscript has been judged scientifically suitable for publication and will be formally accepted for publication once it meets all outstanding technical requirements.

Kind regards,

Hidenori Yamasue, M.D., Ph.D.

Academic Editor

PLOS ONE
---

## [Editor Report · Acceptance letter]

24 Aug 2023

PONE-D-23-06771R2 

Is serotonin transporter brain binding associated with the cortisol awakening response? An independent non-replication 

Dear Dr. Frokjaer:

I'm pleased to inform you that your manuscript has been deemed suitable for publication in PLOS ONE. Congratulations! Your manuscript is now with our production department. 

Kind regards, 

on behalf of

Dr. Hidenori Yamasue 

Academic Editor

PLOS ONE